# Forget About the LiDAR: Self-Supervised Depth Estimators with MED Probability Volumes

**Juan L. Gonzalez**
juanluisgb@kaist.ac.kr

**Munchurl Kim**
mkimee@kaist.ac.kr

School of Electrical Engineering
Korea Advanced Institute of Science and Technology

## Abstract

Self-supervised depth estimators have recently shown results comparable to the supervised methods on the challenging single image depth estimation (SIDE) task, by exploiting the geometrical relations between target and reference views in the training data. However, previous methods usually learn forward or backward image synthesis, but not depth estimation, as they cannot effectively neglect occlusions between the target and the reference images. Previous works rely on rigid photometric assumptions or on the SIDE network to infer depth and occlusions, resulting in limited performance. On the other hand, we propose a method to "Forget About the LiDAR" (FAL), with Mirrored Exponential Disparity (MED) probability volumes for the training of monocular depth estimators from stereo images. Our MED representation allows us to obtain geometrically inspired occlusion maps with our novel Mirrored Occlusion Module (MOM), which does not impose a learning burden on our FAL-net. Contrary to the previous methods that learn SIDE from stereo pairs by regressing disparity in the linear space, our FAL-net regresses disparity by binning it into the exponential space, which allows for better detection of distant and nearby objects. We define a two-step training strategy for our FAL-net: It is first trained for view synthesis and then fine-tuned for depth estimation with our MOM. Our FAL-net is remarkably light-weight and outperforms the previous state-of-the-art methods with $8\times$ fewer parameters and $3\times$ faster inference speeds on the challenging KITTI dataset. We present extensive experimental results on the KITTI, CityScapes, and Make3D datasets to verify our method's effectiveness. To the authors' best knowledge, the presented method performs the best among all the previous self-supervised methods until now.

## 1 Introduction

Single Image Depth Estimation (SIDE) is a critical computer vision task that has been pushed forward by the recent advances in deep convolutional neural networks (DCNNs). In particular, the self-supervised SIDE methods, which exploit geometrical dependencies in the training data, have shown promising results [11, 12, 31], even compared to those of the methods that are supervised with depth ground-truth [2, 3, 17, 30]. However, the previous self-supervised SIDE methods [11, 12, 31] fail because they are not trained directly for depth estimation, but indirectly for view synthesis. In these methods, the occluded regions among the training images prevent them from learning precise depth.

We present a self-supervised method that can accurately learn the SIDE with our novel Mirrored Exponential Disparity (MED) probability volumes. We show that our self-supervised SIDE method achieves superior performance than the state-of-the-art (SOTA) self-, semi- and fully-supervised meth-

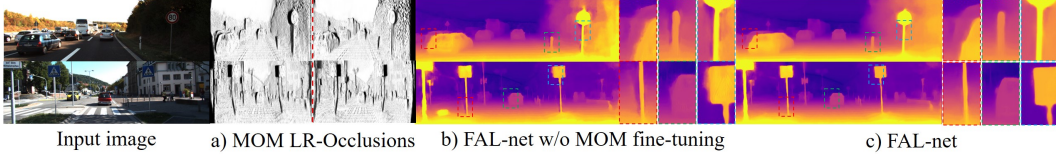

| Input image | a) MOM LR-Occlusions | b) FAL-net w/o MOM fine-tuning | c) FAL-net |

Figure 1: Our proposed FAL-net with and without our novel Mirror Occlusion Module (MOM).

ods on the challenging KITTI [5] dataset. Hence, we propose to "Forget About the LiDAR"(FAL), or 3D laser scanning, for the supervised training of SIDE DCNNs. We recognize that instead of focusing our efforts on developing unnecessary complex (and large) DCNN architectures, it is more worthwhile to focus on loss functions and training strategies that can better exploit the geometrical dependencies in the data for effective self-supervision. Our network, which we call FAL-net, incorporates our proposed MED Probability Volumes into SIDE and achieves higher performance than all the most recent SOTA methods of [11, 12], with almost $8\times$ fewer model parameters. Moreover, our proposed method performs inference of full-resolution depth maps more than $3\times$ faster than [11, 12]. The main contributions of our work are summarized as follows:

1. A novel **Mirrored Occlusion Module (MOM)**, which is a multi-view occlusion mask generation module. The generated masks are very realistic and are used to filter the invalid image regions due to parallax for two images with known (or estimated) camera positions (see Fig.1-(a)).
2. A new **two-stage training strategy**: Firstly, we train our FAL-net for stereoscopic view synthesis penalizing the synthetic right-view in all image regions (see Fig.1-(b)); Secondly, we train our FAL-net for SIDE using our MOM to remove the burden of learning the synthesis of right-occluded contents which are not related to depth, and to provide self-supervision signals for the left-occluded regions which are ignored in the photometric reconstructions (see Fig.1-(c)).
3. We shed light on the effectiveness of **Mirrored Exponential Disparity (MED)** representations for self-supervised SIDE. This small change from the linear to the exponential domain makes our FAL-net, even without MOM, perform surprisingly well, compared to the current SOTA methods.

In the following section, we quickly review the most recent related works, followed by our method in Section 3, and our experimental results in Section 4. We conclude our work in Section 5.

## 2 Related Works

Many recent works have tackled the SIDE task. These can be divided into supervised methods [2, 3, 15, 17, 30], which use the hard-to-obtain depth ground-truth, and the self-supervised methods, which usually learn SIDE from left-right (LR) stereo pairs [7–9, 20, 22, 26, 31, 32] or video [6, 11, 12, 34, 35].

### 2.1 Supervised Methods

Among the top-performing fully-supervised SIDE methods, we can find the works of [3, 17]. Fu *et al.* in their DORN [3] proposed to learn SIDE not as a regression task, but as a classification task by discretizing depth predictions and ground-truths (GTs) in $N$ intervals (quantized levels). On the other hand, Luo *et al.* [17] proposed to train a SIDE network with both depth GT and stereo pairs. They first synthesized a right-view from a left-view with a Deep3D-like [33] network, and then, similar to [18], trained a stereo matching network in a fully-supervised manner.

### 2.2 Unsupervised Methods

Learning to predict depth without labels is a powerful concept, as the commodity cameras are not limited by resolution nor distance as much as the expensive LiDAR equipment. Learning depth in a self-supervised fashion is possible, thanks to the geometrical relationships between multiple captures of the same scene. For the stereo case, some of the most prominent recent works include [22, 26, 31]. For the video case, the works of [10–12] are among the top-performing methods.

For the stereo case, Poggi *et al.* [22] proposed learning from a trinocular setup with center (C), left (L), and right (R) views. Their 3Net [22] is trained with an interleaved protocol and has two decoders to produce C↔L and C↔R respectively. During inference, the final center disparity is obtained by

combining CL and CR disparities following the post-processing in [7]. On the other hand, the recent works of Tosi *et al.* [26] and Watson *et al.* [31] explored incorporating classical stereo-disparity estimation techniques, such as semi-global matching (SGM), as additional supervision for the SIDE task. In these works, the SGM proxy labels are distilled either by LR-consistency checks [26] or by analyzing the photometric reconstruction errors [31].

For the monocular video case, the work of Gordon *et al.* [10] proposed to learn not only camera pose and depth (in a similar way as in the early work of Zhou *et al.* [35]), but also camera intrinsics and lens distortion. Additionally, a segmentation network was used to predict and potentially ignore likely-moving objects (truck, bike, car, etc.), as these do not contribute to the learning process. On the other hand, Guizilini *et al.* proposed Pack-net [11], which is a powerful auto-encoder network with ~120M parameters, and 3D packing and unpacking modules. These modules utilize sub-pixel convolutions and deconvolutions [24] instead of striding or pooling, and process the resulting channels with standard 3D and 2D convolutions. Learning from video is carried out in a similar way to [6], with an optional velocity supervision loss for scale-aware structure-from-motion. Supported by a pre-trained semantic segmentation network and a PackNet [11] backbone, the later work of Guizilini *et al.* [12] also learns from videos. Their method injects segmentation-task features into the decoder side of their network, which helps to generate structurally better predictions.

## 3 Method

It has been shown that DCNNs can learn to predict depth from a single image in a self-supervised manner when two or more views with known (or estimated) camera positions are available [4, 6–9, 11, 20, 22, 32, 35]. Learning is commonly carried out by minimizing reconstruction errors between depth-guided synthesized images and available views. However, reducing such an objective loss function involves estimating the contents in the occluded regions, which degrades the networks' performance on the depth estimation task. Previous works have attempted to handle such occluded areas by learning uncertainty masks [8, 35], analyzing the photometric reconstruction errors during training time [6, 10–12, 32], and by letting the network hallucinate the occluded regions in the image synthesis task [9, 33]. All these methods fail in effectively making the occluded regions transparent to the networks, as the geometrical dependencies of the given views are not taken into account. Moreover, these methods become overloaded with the task of generating such uncertainty masks or occluded contents, thus leading to the waste of their learning capacities for depth estimation during training time. To solve this issue, our FAL-net with Mirrored Exponential Disparity (MED) probability volumes and our new two-step training strategy are proposed.

### 3.1 Network Architecture

Before delving into our training strategy, it is worth to review our simple, yet effective FAL-net architecture with Mirrored Exponential Disparity (MED) probability volumes. The FAL-Net is a 6-stage auto-encoder with one residual block after each strided convolution stage in the encoder side and skip-connections between the encoder and the decoder. More details can be found in the Supplemental. Our FAL-net maps a single left-view image $\mathbf{I}_L$ to a $N$-channel disparity logit volume $\mathbf{D}_L^L$, $f : \mathbf{I}_L \mapsto \mathbf{D}_L^L$. $\mathbf{D}_L^L$ can be passed through a softmax operation along the channel axis to obtain the left-view MED probability volume $\mathbf{D}_L^{PL}$. A sum of the $N$ channels of $\mathbf{D}_L^{PL}$, weighted by the exponential disparity level value $d_n$, reveals the final predicted disparity map $\mathbf{D}_L'$, as given by

$$\mathbf{D}_L' = \sum_{n=0}^{N} d_n \mathbf{D}_{L_n}^{PL} \qquad (1) \qquad\qquad d_n = d_{max} e^{\ln \frac{d_{max}}{d_{min}} \left( \frac{n}{N} - 1 \right)} \qquad (2)$$

where $d_{min}$ and $d_{max}$ are the minimum and maximum disparity hyper-parameters respectively. Each $n$-channel of $\mathbf{D}_L^L$ can be warped (shifted) into the right view camera by the warping operation $g(\cdot, d_n)_{L \to R}$, and the resulting $N$-channel stack soft-maxed along the channel axis to obtain the right-from-left MED probability volume $\mathbf{D}_L^{PR}$. The element-wise multiplication, denoted by $\odot$, of $\mathbf{D}_L^{PR}$ with equally warped $N$ versions of $\mathbf{I}_L$, followed by a *sum-reduction* operation, produces a synthetic right view $\mathbf{I}_R'$. This process is shown in the top-left of Fig. 3 and is described by

$$\mathbf{I}_R' = \sum_{n=0}^{N} g\left(\mathbf{I}_L, d_n\right)_{L \to R} \odot \mathbf{D}_{L_n}^{PR} \qquad (3)$$

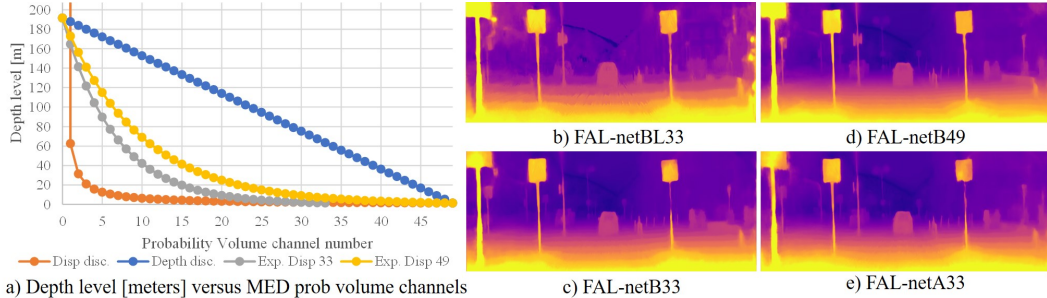

a) Depth level [meters] versus MED prob volume channels

b) FAL-netBL33

c) FAL-netB33

d) FAL-netB49

e) FAL-netA33

Figure 2: Effects of exponential disparity discretization.

The resulting left view depth $\mathbf{D}'_L$ and synthetic right view $\mathbf{I}'_R$ can be used to train the FAL-net in a self-supervised fashion. Still, more importantly, the MED probability volumes ($\mathbf{D}^{PL}_L$, $\mathbf{D}^{PR}_L$, $\mathbf{D}^{PR}_R$, and $\mathbf{D}^{PL}_R$) can be used in our novel Mirrored Occlusion Module (MOM) for accurate SIDE learning.

### 3.1.1 Exponential Disparity Discretization

Disparity probability volumes can be understood as the kernel elements of adaptive convolutions [9] or as a way of depth discretization [3]. When interpreted as adaptive convolutions for the task of new view synthesis as in [9, 17, 33], it might be reasonable to use linear quantization disparity levels as they produce equally spaced kernel sampling positions. However, due to the inverse relation between disparity and depth, linear quantization of disparity implies that most sampling positions will be used for the very close-by objects, as depicted with the orange curve in Figure 2-(a). Linear quantization in depth units is also not adequate, as it assigns very few sampling positions for the very close-by objects as depicted with the blue curve in Fig. 2-(a). In a similar spirit to [3], we propose exponential disparity discretization, which is described by Eq. 2 and depicted in Figure 2-(a) in yellow and gray curves for $N = 49$ and $N = 33$ levels respectively. The effect of training our FAL-net with 33-linear, 33-exponential, and 49-exponential disparity quantization levels is shown in Fig. 2-(b,c,d) respectively. In contrast with [3] our MED probability volumes are allowed to take any value from 0 to 1 (guided by the channel-wise softmax), as we do not impose a one-hot encoding classification loss. This freedom allows our MED probability volumes to softly blend when computing the final disparity map, which helps in obtaining higher accuracy than [3] with fewer quantization levels.

## 3.2 Training Strategy

We define a two-step training strategy. In the first step, we train our FAL-net for view synthesis with $l1$, perceptual [14], and smoothness losses, and keep a fixed copy of the trained model. In the second step, enabled by our Mirrored Occlusion Module, we fine-tune our FAL-net for inverse depth (disparity) estimation with an occlusion-free reconstruction loss, smoothness loss, and a "mirror loss". Our mirror loss uses a mirrored disparity prediction $\mathbf{D}'_{LM}$, generated by the fixed model, to provide self-supervision only to the regions that are occluded in the right view but visible in the left view.

### 3.2.1 Mirrored Occlusion Module

Our novel Mirrored Occlusion Module (MOM) is a multi-view occlusion mask generation module which allows our FAL-net to directly learn SIDE by cross-generating occlusion maps from the MED probability distributions of two training images with known (or estimated) camera positions. These generated occlusion maps get improved as the network learns better depth. Our second training step with the MOM is depicted in Figure 3. At each iteration, the FAL-net runs the forward pass for each left and right view to obtain the MED probability volumes $\mathbf{D}^{PL}_L$, $\mathbf{D}^{PR}_L$, $\mathbf{D}^{PR}_R$, and $\mathbf{D}^{PL}_R$. In our MOM, all the channels of each probability volume are warped to their opposite camera views by $g(\cdot, d_n)$ correspondingly, and reduced to one single channel via summation to give rise to four sub-occlusion masks (two for each input view). The sub-occlusion masks that are aligned to their respective input view are more detailed, and can be further refined by an element-wise multiplication with their homologous (or "mirror") sub-occlusion generated by the opposite view, as can be observed in the bottom of Fig. 3. These operations yield the two final occlusion masks $\mathbf{O}^L$ and $\mathbf{O}^R$. The occlusion

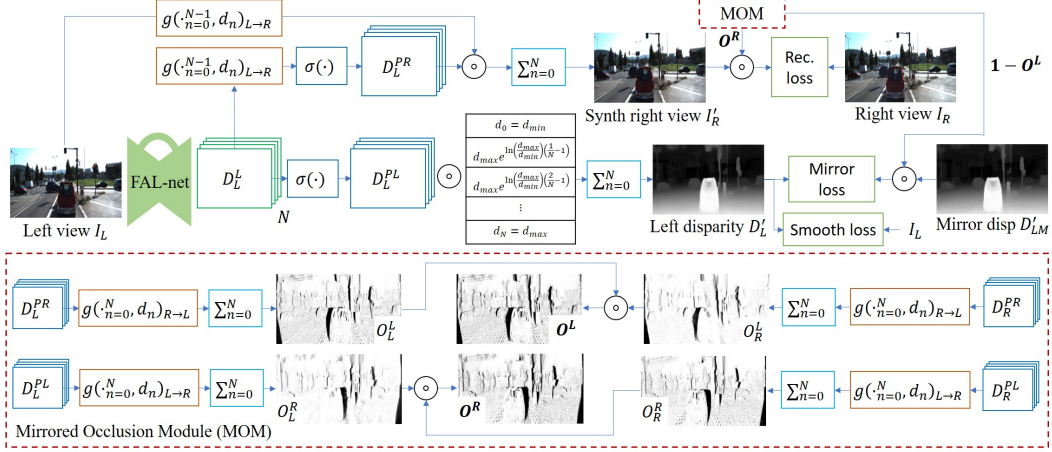

Figure 3: Our proposed training strategy and novel Mirrored Occlusion Module (MOM).

masks can be given as a function of $\mathbf{D}_L^{PL}$ and $\mathbf{D}_R^{PL}$ by ($\mathbf{O}^L$ is described by swapping R↔L)

$$\mathbf{O}^R = \max \left[ \sum_{n=0}^{N} g\left(\mathbf{D}_{L_n}^{PL}, d_n\right)_{L \to R} \right] \odot \left[ \sum_{n=0}^{N} g\left(\mathbf{D}_{R_n}^{PL}, d_n\right)_{L \to R} \right], 1 \tag{4}$$

Please note that the sum-reduction operation on the sub-occlusion masks is not bounded between 0 and 1, as the planes of the probability distributions are first warped (shifted) to the target view by $g(\cdot)$ in Eq. (4). This shifting not only generates "holes", which are the occluded regions, but also areas where the summation is $> 1$. The latter is the reason why the "max" operator (not shown in Fig. 3) is applied to cap the final occlusion masks $O^L$ and $O^R$ between 0 and 1.

Also note that $\mathbf{O}^L$ and $\mathbf{O}^R$ are both needed for training on depth estimation. For a left input view $\mathbf{I}_L$, $\mathbf{O}^R$ is used to prevent the network from learning view synthesis, as $\mathbf{O}^R$ effectively removes right-occluded contents that are only visible in the right view. On the other hand, $\mathbf{O}^L$, in combination with a mirrored disparity estimate $\mathbf{D}'_{LM}$, is used to provide self-supervision signals to the output disparity values corresponding to the left-occluded regions that are only visible in the left view, as shown in the top-right of Fig. 3. The dark regions in $\mathbf{O}^L$ (not visible in the right view) are not affected by the photometric reconstruction losses and often result in depth artifacts, as can be observed in Fig. 1-(b). $\mathbf{D}'_{LM}$ is obtained by feeding the fixed FAL-net with a horizontally flipped version of $\mathbf{I}_L$, and flipping the output disparity again. It is well-known that this operation is equivalent to making the network treat $\mathbf{I}_L$ as the right view, thus generating artifacts on the right-side instead of the left-side of the objects [7, 9, 20, 22]. Note that contrary to [20, 22], the contribution of the fixed FAL-net through $\mathbf{D}'_{LM}$ is weighted by $1 - \mathbf{O}_L$, which prevents the FAL-net under training from learning over-smoothness.

### 3.2.2 Loss Functions

The total loss for learning inverse depth, as used in the second step of training, is given by:

$$l = \frac{1}{2}(l_{rec}^L + l_{rec}^R + l_m^L + l_m^R + \alpha_{ds}l_{ds}^L + \alpha_{ds}l_{ds}^R) \tag{5}$$

where the total loss is divided by 2 as the network runs on the left and right views. $\alpha_{ds}$ weights the contribution of the smoothness loss. $\alpha_{ds}$ was empirically found effective when set to 0.0008 during the first training step and doubled to 0.0016 in the second step of MOM fine-tuning.

**Occlusion-free reconstruction loss.** The combination of $l1$ and perceptual loss [14] has shown to be effective for multiple tasks that involve image reconstruction or view synthesis [9, 19, 36]. We adopt this combination to enforce a similarity between the training views and their synthetic counterparts. The first 3 maxpool layers, denoted by $\phi^l(\cdot)$, from the pre-trained VGG19 [25] on the ImageNet classification task were used on our occlusion-free reconstruction loss, which is given by

$$l_{rec}^R = ||\mathbf{O}^R \odot (\mathbf{I}'_R - \mathbf{I}_R)||_1 + \alpha_p \sum_{l=1}^{3} ||\phi^l(\mathbf{O}^R \odot \mathbf{I}'_R + (1 - \mathbf{O}^R) \odot \mathbf{I}_R) - \phi^l(\mathbf{I}_R)||_2^2 \tag{6}$$

Table 1: Ablation studies on KITTI [5] (K). CS→K: Trained on CS and re-trained on K. K+CS: Concurrent K and CS training. #Par: Parameters in millions. K+20e: fine tuned with +20 epochs

| # | Methods | data | #Par | abs rel↓ | sq rel↓ | rmse↓ | $rmse_{log}$ ↓ | $a^1$ ↑ | $a^2$ ↑ | $a^3$ ↑ |
|---|---------|------|------|----------|---------|-------|----------------|---------|---------|---------|
| 7 | FAL-netB49 | K+CS | 17 | **0.071** | **0.287** | 2.905 | **0.109** | **0.941** | **0.990** | **0.998** |
| | FAL-netB49 | K | 17 | 0.075 | 0.298 | 2.905 | 0.112 | 0.937 | 0.989 | 0.997 |
| | FAL-netB33 | K | 17 | 0.076 | 0.304 | **2.890** | 0.112 | 0.938 | 0.989 | 0.997 |
| | FAL-netA33 | K | 6.6 | 0.085 | 0.367 | 3.161 | 0.124 | 0.924 | 0.986 | 0.997 |
| | FAL-netB49 | CS | 17 | 0.112 | 0.559 | 3.950 | 0.158 | 0.876 | 0.974 | 0.993 |
| 6 | FAL-netB33 (scratch) | K | 17 | 0.078 | 0.330 | 2.950 | 0.113 | 0.938 | 0.989 | 0.997 |
| 5 | FAL-netB33 w/o MOM | K+20e | 17 | 0.081 | 0.349 | 3.259 | 0.120 | 0.928 | 0.987 | 0.997 |
| 4 | FAL-netB49 w/o MOM | K+CS | 17 | 0.074 | 0.318 | 3.086 | 0.114 | 0.935 | 0.989 | 0.997 |
| | FAL-netB49 w/o MOM | CS→K | 17 | 0.085 | 0.391 | 3.229 | 0.125 | 0.924 | 0.985 | 0.996 |
| | FAL-netB49 w/o MOM | CS | 17 | 0.127 | 0.721 | 4.406 | 0.179 | 0.845 | 0.961 | 0.988 |
| 3 | FAL-netB49 w/o MOM | K | 17 | 0.076 | 0.331 | 3.167 | 0.116 | 0.932 | 0.988 | 0.997 |
| 2 | FAL-netA33 w/o MOM | K | 6.6 | 0.087 | 0.386 | 3.303 | 0.127 | 0.921 | 0.986 | 0.997 |
| | FAL-netB33 w/o MOM | K | 17 | 0.079 | 0.329 | 3.033 | 0.116 | 0.933 | 0.988 | 0.997 |
| | FAL-netC33 w/o MOM | K | 26 | 0.080 | 0.344 | 3.184 | 0.119 | 0.928 | 0.987 | 0.997 |
| 1 | FAL-netBL33 w/o MOM | K | 17 | 0.109 | 0.890 | 6.118 | 0.190 | 0.845 | 0.950 | 0.982 |

where $\odot$ is the hadamard product. Note that $\mathbf{O}^R$ blends $\mathbf{I}'_R$ with $\mathbf{I}_R$ to be fed "occlusions-free" to the VGG19. $\alpha_p$ roughly balanced the contribution between the $l1$ and perceptual losses and was empirically set to $\alpha_p$=0.01 for all our experiments. See Supplemental for results on $\alpha_p$=0. Setting $\mathbf{O}^R$=$\mathbf{O}^L$=1 yields the vanilla reconstruction loss used in the first step of learning view synthesis.

**Edge-preserving smoothness loss.** We adopt the widely used edge-preserving smoothness loss [7, 8, 11]. In our FAL-net, this term prevents the network from learning depth distributions that would give rise to "too much occlusions" in our MOM. In contrast with the previous works, we add a parameter $\gamma = 2$ to regulate the amount of edge preservation.

$$l_{ds}^L = ||\partial_x \mathbf{D}'_L \odot e^{-\gamma|\partial_x \mathbf{I}_L|}||_1 + ||\partial_y \mathbf{D}'_L \odot e^{-\gamma|\partial_y \mathbf{I}_L|}||_1 \qquad (7)$$

**Mirror loss.** This term provides self-supervision signals to the visible contents in the left view that are occluded in the right view, from a pass of the fixed FAL-net on a mirrored $\mathbf{I}_L$. As can be noted from $\mathbf{O}^L$ in Fig. 3, the contribution of the mirrored disparity map $\mathbf{D}'_{LM}$ is very limited, and given by

$$l_m^L = (1/max(\mathbf{D}'_{LM}))||(1 - \mathbf{O}^L) \odot (\mathbf{D}'_L - \mathbf{D}'_{LM})||_1 \qquad (8)$$

where $max(\mathbf{D}'_{LM})$ is the maximum disparity value in $\mathbf{D}'_{LM}$ that weights down the mirror loss.

## 4 Experiments and Results

Experiments were mainly conducted on the benchmark dataset, KITTI [5], which contains stereo images captured from a driving perspective with projected 3D laser scanner pseudo-ground-truths. For training, the widely used Eigen train split [2] was adopted, which consists of 22,600 training LR pairs. Using the metrics defined in [2], we evaluated our models on the two Eigen test split datasets: the original [2] and the improved versions [28], consisting of 697 and 652 test images respectively. We also trained and evaluated our FAL-net on the high-resolution CityScapes [1] driving dataset and on the Make3D [23] dataset, respectively, to challenge the generalization power of our method.

### 4.1 Implementation Details

All our models were trained with the Adam optimizer with default betas with a batch size of 8 (4 left, 4 right). In the first training step (view synthesis), our FAL-net was updated via 50 epochs with an initial learning rate (lr) of $1 \times 10^{-4}$ that halves at epochs 30 and 40. In the second training step (depth estimation), our model is trained for 20 epochs, with an initial lr of $5 \times 10^{-5}$, that halves at epoch 10. Random data augmentations are performed on-the-fly with resize (with a factor of 0.75 to 1.5) followed by a crop of size 192x640, horizontal flipping, and changes in gamma, brightness, and color brightness. Training takes 3 days for the first step and 1 day for the second step on a Titan XP GPU.

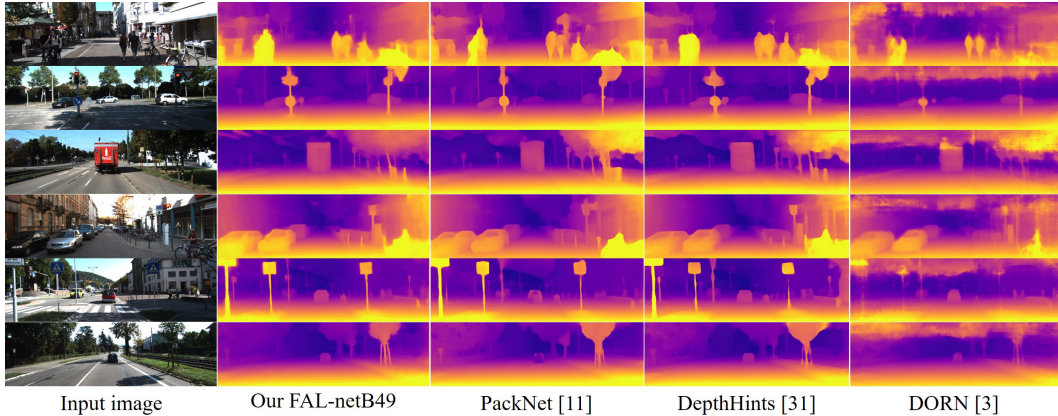

| Input image | Our FAL-netB49 | PackNet [11] | DepthHints [31] | DORN [3] |

Figure 4: Qualitative comparison among recent SIDE methods. Our predictions are more detailed.

## 4.2 Ablation Studies

We present an extensive ablation study in Table 1 for our FAL-net trained on the KITTI Eigen train split [2] (K) and tested on the improved KITTI Eigen test split [28]. Table 1 shows from bottom to top: **#1** Regressing disparity in the linear space yields very poor results; **#2** The effect of the numbers of parameters is shown. Our FAL-net without MOM fine-tuning seems to achieve its performance with the relatively few parameters (17M) of the FAL-netB33; **#3** The number of quantized disparity levels is ablated, which surprisingly does not show a considerable difference, at least in quantitative metrics. Qualitatively, it is shown in Fig. 2 that the use of 49 levels yields smoother predictions; **#4** We explored the traditional CityScapes (CS) pre-training [7] and the concurrent K+CS training [9, 10]. Higher performance was obtained for K+CS; **#5** We trained a FAL-netB33 without MOM for additional 20 epochs, which did not bring improved performance. This implies that the performance gain obtained from fine-tuning with MOM in the second training step does not come from the additional epochs; **#6** The use of our MOM is evaluated, but instead of fine-tuning, we only keep the pre-trained weights for the fixed model and train the FAL-net from "scratch" for 50 epochs. This yielded the same level of improvement as fine-tuning in (#7), but with the additional training time; **#7** Lastly, we fine-tuned with MOM and observed consistent gains across all configurations. The models trained with CS and K+CS also presented consistent improvements in all metrics, which indicate that the gains from MOM fine-tuning are not tied to a specific dataset.

Additionally, to further show that the effectiveness of the proposed "exponential disparity level" representation is universal, we plugged MED and MOM into different network backbones, the Monodepth [7] and the more recent SuperDepth [20]. The first has a simpler but heavier auto-encoder backbone than our FAL-net, and the latter incorporates ESPCN [24] up-sampling modules in the decoder stage. Their counterparts with MED representations are denoted in Table 3 as Monodepth-MED1 and Superdepth-MED1, for the models in the first training stage; and as Monodepth-MED2 and Superdepth-MED2, for the models with MOM fine-tuning (second training stage). Incorporating MED volumes and Fine-tuning with MOM showed steady improvements in both networks, which further supports our overall method's effectiveness.

## 4.3 Results

**KITTI.** Table 3 shows the performance of our method in comparison with the best results of the previous SOTA. Our method performs favorably versus all previous fully- and self-supervised methods, achieving the best results on the majority of the metrics in the original [2] and improved [28] Eigen test splits. Results from [6, 11, 31] were obtained from their publicly available repositories. Fig. 4 shows that our FAL-net, even with much fewer parameters than [11, 12], infers depth of thin and complex structures more consistently. Note that, in contrast with [6, 11, 12, 20] that obtain their best results by training on full-resolution images, our method is trained on a 192×640 patch and evaluated on the full-resolution 384×1280 image, on which our method performs inference in 19ms, more than 3× faster than the SOTA of [11, 12]. Additionally, It is remarkable that our FAL-net with a low parameter count outperforms previous methods that exploit other supervision signals, such as Depth, SGM, or semantics. For a fairer comparison with [22, 26, 31], we adopt a post-processing step (PP),

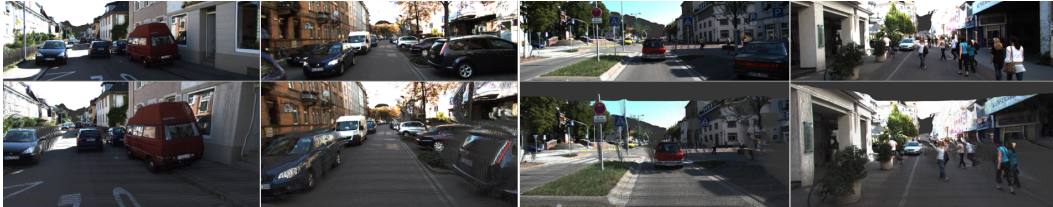

Figure 5: Textured point-clouds generated by our FAL-netB49 network predictions. Top row: Input image, bottom row: point-cloud from different view points ($[0\text{m}, 0.2\text{m}, 3\text{m}, -11°, 0°, 0°]$ left two columns, $[-0.9\text{m}, 0\text{m}, 2\text{m}, 0°, 0°, 0°]$ right two columns).

Table 2: Results on Make3D [23]. All self-supervised methods benefit from median scaling. Evaluations with C1 [16] metrics (up to 70m). M3D: Train on the Make3D [23] dataset

| Method | Sup | Data | abs rel | sq rel | rmse | Method | Sup | Data | abs rel | sq rel | rmse |
|---|---|---|---|---|---|---|---|---|---|---|---|
| Liu *et al.* [16] | D | M3D | 0.475 | 6.562 | 10.05 | Laina *et al.* [15] | D | M3D | **0.204** | **1.840** | **5.683** |
| SFMLearner [35] | V | K | 0.383 | 5.321 | 10.47 | Monodepth (PP) [7] | S | CS | 0.443 | 7.112 | 8.860 |
| Monodepth2 [6] | V | K | 0.322 | 3.589 | 7.417 | Wang *et al.* [29] | S | K | 0.387 | 4.720 | 8.090 |
| Zhou *et al.* [34] | V | K | 0.318 | 2.288 | 6.669 | Glez. and Kim [8] | S | K | 0.323 | 4.021 | 7.507 |
| FAL-netB49 | S | K | 0.297 | 2.913 | 6.810 | FAL-netB49 | S | K+CS | 0.256 | 2.179 | 6.201 |
| FAL-netB49 (PP) | S | K | 0.284 | 2.803 | 6.643 | FAL-netB49 (PP) | S | K+CS | **0.254** | **2.140** | **6.139** |

but instead of following [7], we implement a less-expensive multi-scale PP which further improves our performance. More details on our PP in Supplemental. Results on the improved Eigen split [28] without PP are not shown in Table 3, as they are already provided in Table 1. Note that our method without PP already outperforms the SOTA in most metrics, specially sq rel [2] and RMSE. To further assess the stability and metric alignment of our depth estimates, we present textured point-clouds seen from novel camera view-points in Figure 5. It is worth noting that our proposed method can generate accurate point-clouds, which are consistent when seen from camera positions (for example, 3m in Figure 5) that even exceed the camera baseline in the training dataset (0.54m).

**CityScapes.** The use of CityScapes [1] (CS) demonstrates our method can generalize well. Table 3 shows our model trained on CS only achieves better performance when evaluated on KITTI than [10]. Our method can also leverage more training data, as shown by our method trained with K+CS.

**Make3D.** Table 2 shows the performance of our FAL-net evaluated on the Make3D [23] dataset following the protocol in [7]. Our method, not fine-tuned on Make3D, achieves the best performance versus other self-supervised methods. More results available on the Supplementary materials.

# 5 Discussion

It was shown in Section 4 that when our network is fine-tuned with MOM, it can gradually correct mistakes in occlusion and depth estimation, leading to considerable improvements versus the model only with one stage of training. However, this leads to a question, if the computed occlusions in the MOM are not accurate to begin with, how can the FAL-net get better? Please note that the MOM is fed with the MED volumes from the FAL-net under training, not from the fixed FAL-net model from the first training stage. This means that the generated occlusion maps in the MOM will get better as the depth estimates from the FAL-net under training get better. In addition, the MOM obtains the occlusion masks for each view from both views. That is, if one view fails to obtain an accurate sub-occlusion map, the final occlusion mask has the chance to be fixed with the corresponding sub-occlusion mask obtained from the opposite view.

Suppose the occlusion masks obtained from the MOM are incorrect. In the worst-case scenario, the total loss function will become either the same as that in the first training stage (with $\mathbf{O}^R$ mask all-ones) or will have its reconstruction term in Eq. 6 suppressed (with $\mathbf{O}^R$ mask all-zeros and $\mathbf{O}^L$ mask all-ones). The latter will make the total loss function not convey any self-supervision signal other than the flipped network output in the mirror loss of Eq. 8. This suggests that the network cannot get worse than its initial state when fine-tuning with the MOM. Our FAL-net can only get better until a certain point, which is also limited by the mirror disparity estimate from the fixed FAL-net $\mathbf{D}'_{LM}$.

Table 3: Performance comparison of existing SIDE methods. K: Eigen [2] train-split. CS: CityScapes [1]. PP: post-processing. CS→K: CS pre-training. K+CS: Concurrent K and CS training. DoF and D: Depth-of-field and depth supervision. S, $S_{SGM}$, V, V+v, V+Se: Stereo, stereo+SGM, video, video + velocity, and video + semantics self-supervision. V methods benefit from median-scaling. LR: eval at low-resolution. ↓↑ indicate the better metric. **Best** and second-best metrics. Results capped to 80m

| Ref | Methods | PP | Sup | Data | #Par | abs rel↓ | sq rel↓ | rmse↓ | rmse$_{log}$↓ | $a^1$↑ | $a^2$↑ | $a^3$↑ |
|---|---|---|---|---|---|---|---|---|---|---|---|---|
| | | | | | | Original Eigen Test Split [2] | | | | | | |
| [13] | Gur *et al.* | | DoF | K | - | 0.110 | 0.666 | 4.186 | **0.168** | 0.880 | 0.966 | **0.988** |
| [17] | Luo *et al.* | | D+S | K | - | 0.094 | 0.626 | 4.252 | 0.177 | 0.891 | 0.965 | 0.984 |
| [10] | Gordon *et al.* | | V | K | - | 0.128 | 0.959 | 5.230 | 0.212 | 0.845 | 0.947 | 0.976 |
| [34] | Zhou *et al.* | | V | K | 34 | 0.121 | 0.837 | 4.945 | 0.197 | 0.853 | 0.955 | 0.982 |
| [6] | Monodepth2 | | V | K | 14 | 0.115 | 0.882 | 4.701 | 0.190 | 0.879 | 0.961 | 0.982 |
| [11] | PackNet | | V | K | 120 | 0.107 | 0.802 | 4.538 | 0.186 | 0.889 | 0.962 | 0.981 |
| [10] | Gordon *et al.* | | V | K+CS | - | 0.124 | 0.930 | 5.120 | 0.206 | 0.851 | 0.950 | 0.978 |
| [11] | PackNet | | V | CS→K | 120 | 0.104 | 0.758 | 4.386 | 0.182 | 0.895 | 0.964 | 0.982 |
| [11] | PackNet | | V+v | CS→K | 120 | 0.103 | 0.796 | 4.404 | 0.189 | 0.881 | 0.959 | 0.980 |
| [12] | Guizilini *et al.* | | V+Se | CS→K | 140 | 0.100 | 0.761 | 4.270 | 0.175 | **0.902** | 0.965 | 0.982 |
| [7] | Monodepth | | S | K | 32 | 0.148 | 1.344 | 5.927 | 0.247 | 0.803 | 0.922 | 0.964 |
| [20] | SuperDepth | | S | K | - | 0.112 | 0.875 | 4.958 | 0.207 | 0.852 | 0.947 | 0.977 |
| [26] | Tosi *et al.* | ✔ | $S_{SGM}$ | K | 42 | 0.111 | 0.867 | 4.714 | 0.199 | 0.864 | 0.954 | 0.979 |
| [21] | Refine&Distill | | S | K | - | 0.098 | 0.831 | 4.656 | 0.202 | 0.882 | 0.948 | 0.973 |
| [31] | DepthHints | ✔ | $S_{SGM}$ | K | 35 | 0.096 | 0.710 | 4.393 | 0.185 | 0.890 | 0.962 | 0.981 |
| [22] | 3Net | ✔ | S | CS→K | 48 | 0.111 | 0.849 | 4.822 | 0.202 | 0.865 | 0.952 | 0.978 |
| [26] | Tosi *et al.* | ✔ | $S_{SGM}$ | CS→K | 42 | 0.096 | 0.673 | 4.351 | 0.184 | 0.890 | 0.961 | 0.981 |
| [7] | Monodepth-MED1 | | S | K | 32 | 0.112 | 0.751 | 4.500 | 0.196 | 0.868 | 0.954 | 0.980 |
| [7] | Monodepth-MED2 | | S | K | 32 | *0.107* | *0.684* | *4.311* | *0.187* | *0.878* | *0.960* | *0.982* |
| [20] | SuperDepth-MED1 | | S | K | 58 | 0.111 | 0.682 | 4.295 | 0.190 | 0.879 | 0.959 | 0.982 |
| [20] | SuperDepth-MED2 | | S | K | 58 | *0.108* | *0.647* | *4.180* | *0.184* | *0.886* | *0.962* | *0.983* |
| our | FAL-netB33 | | S | K | 17 | 0.099 | 0.633 | 4.074 | 0.177 | 0.894 | 0.965 | 0.984 |
| our | FAL-netB33 | ✔ | S | K | 17 | 0.094 | 0.597 | 4.005 | 0.173 | 0.900 | **0.967** | 0.985 |
| our | FAL-netB49 | | S | K | 17 | 0.097 | 0.590 | 3.991 | 0.177 | 0.893 | 0.966 | 0.984 |
| our | FAL-netB49 | ✔ | S | K | 17 | 0.093 | 0.564 | **3.973** | 0.174 | 0.898 | **0.967** | 0.985 |
| our | FAL-netB49 | | S | K+CS | 17 | 0.091 | 0.562 | 4.016 | 0.178 | 0.894 | 0.964 | 0.983 |
| our | FAL-netB49 | ✔ | S | K+CS | 17 | **0.088** | **0.547** | 4.004 | 0.175 | 0.898 | 0.966 | 0.984 |
| [10] | Gordon *et al.* | | V | CS | - | 0.172 | 1.370 | 6.210 | 0.250 | 0.754 | 0.921 | 0.967 |
| our | FAL-netB49 | | S | CS | 17 | *0.144* | *0.871* | *4.796* | *0.215* | *0.811* | *0.947* | *0.979* |
| | | | | | | Improved Eigen Test Split [28] | | | | | | |
| [3] | DORN | | D | K | 51 | 0.072 | 0.307 | 2.727 | 0.120 | 0.932 | 0.984 | 0.995 |
| [6] | Monodepth2 | | V | K | 14 | 0.092 | 0.536 | 3.749 | 0.135 | 0.916 | 0.984 | 0.995 |
| [11] | PackNet (LR) | | V | K | 120 | 0.078 | 0.420 | 3.485 | 0.121 | 0.931 | 0.986 | 0.996 |
| [11] | PackNet | | V | CS→K | 120 | 0.071 | 0.359 | 3.153 | 0.109 | **0.944** | 0.990 | 0.997 |
| [11] | PackNet | | V+v | CS→K | 120 | 0.075 | 0.384 | 3.293 | 0.114 | 0.938 | 0.984 | 0.995 |
| [6] | Monodepth2 | | V+S | K | 14 | 0.087 | 0.479 | 3.595 | 0.131 | 0.916 | 0.984 | 0.996 |
| [6] | Monodepth2 | | S | K | 14 | 0.084 | 0.503 | 3.646 | 0.133 | 0.920 | 0.982 | 0.994 |
| [31] | DepthHints | ✔ | $S_{SGM}$ | K | 35 | 0.074 | 0.364 | 3.202 | 0.114 | 0.936 | 0.989 | 0.997 |
| our | FAL-netA33 | ✔ | S | K | 6.6 | 0.076 | 0.335 | 3.122 | 0.116 | 0.934 | 0.989 | 0.997 |
| our | FAL-netB33 | ✔ | S | K | 17 | 0.071 | 0.282 | **2.859** | **0.106** | 0.944 | **0.991** | 0.998 |
| our | FAL-netB49 | ✔ | S | K | 17 | 0.071 | 0.281 | 2.912 | 0.108 | 0.943 | **0.991** | **0.998** |
| our | FAL-netB49 | ✔ | S | K+CS | 17 | **0.068** | **0.276** | 2.906 | **0.106** | **0.944** | **0.991** | **0.998** |

## 6  Conclusion

We have shown that state-of-the-art single image depth estimation (SIDE) can be achieved by light and straightforward auto-encoder networks that incorporate Mirrored Exponential Disparity (MED) probability volumes in their output layers. We showed that a two-step training strategy with our Mirror Occlusion Module (MOM) aids in making the network learn precise depth instead of just view-synthesis. Our method outperforms the DORN [3] supervised baseline by a large accuracy margin and $3\times$ fewer parameters, which suggests we can "forget about the LiDAR" for the supervision of SIDE networks, provided the adequate capture conditions. We hope this work can shift the research efforts towards faster and lighter network architectures for self-supervised SIDE. Moreover, any task that requires proper handling of occluded regions caused by rigid motions such as learning of stereo disparity, SIDE from monocular videos, and optical flow can benefit from our proposed method.

## Broader Impact

In this paper, we presented FAL-net, a method to "forget about the LiDAR" for the learning of monocular depth from stereo images. Our approach incorporates our proposed mirrored exponential disparity (MED) probability volumes and a two-stage learning strategy with our novel mirrored occlusion module (MOM). Our MOM computes very realistic occlusion masks to filter out invalid regions due to parallax. Our FAL-net showed superior performance and reduced number of parameters and inference times than the SOTA fully-, semi-, and self- supervised methods.

Even though we focused on learning single image depth estimation (SIDE) from stereo pairs, our method can be easily extended when learning from monocular videos. Our MOM can be adopted as long as the network incorporates a disparity probability volume in its output layers and the relative camera poses are known or estimated. The camera-pose information can be integrated into the warping operation $g(\cdot)$ in Eq. (4) to obtain the mirrored occlusions for the corresponding frame pair. What could be at stake here is the exponential quantization, as inverse depths in structure-from-motion (SFM) are defined up to an unknown and inconsistent scale. The ambiguous scale could prevent the network from taking advantage of all disparity levels. A turn-around for this issue is to incorporate velocity supervision, as introduced in PackNet [11], or consistent SFM [27] to fully exploit the exponential quantization.

Being depth estimation a low-level computer vision task, we authors do not consider that any ethical implication is involved in our research. However, we believe it is crucial to know if the network consistently under or overestimates depth. The second is considered more critical in robotics systems, in particular, self-driving cars. In this regard, our FAL-net seems to be on the safer side. We measured this by computing the mean median-scaling factor [35] between the GT and our depth estimates. We obtained a mean scale factor of 1.016, indicating that our network detects objects slightly closer than they are.

Finally, we would like to remind the reader that, if one wants to use software-based depth estimators for safety-critical systems, all the necessary redundancy checks and safety norms must be followed.

## Acknowledgments and Disclosure of Funding

This work was supported by Institute for Information & communications Technology Promotion (IITP) grant funded by the Korea government (MSIT) (No. 2017-0-00419, Intelligent High Realistic Visual Processing for Smart Broadcasting Media).

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
