[Supplementary Material]

# Supplementary materials for: Forget About the LiDAR: Self-Supervised Depth Estimators with MED Probability Volumes

**Juan L. Gonzalez**
juanluisgb@kaist.ac.kr

**Munchurl Kim**
mkimee@kaist.ac.kr

School of Electrical Engineering
Korea Advanced Institute of Science and Technology

## 1   Introduction

We present our detailed network architecture, details on our multi-scale post-processing step, and additional analysis and results on the KITTI [4], CityScapes [1] and Make3D [13] datasets in the present supplementary materials. Additionally, we provide animated camera trajectories of four textured point-clouds generated from our FAL-netB49 depth estimates to show our method's accuracy and consistency. These animations are provided as *.gif* files compressed along with these supplementary materials.

## 2   Additional notes on our Mirrored Occlusion Module

In our main paper, we suggest that our Mirrored Occlusion Module (MOM) does not impose a burden on our FAL-net. The reason for this is that our FAL-net does not need to separately infer the occlusion masks, as the MOM can extract them from the MED probability volumes of the given image pairs during training time. To complement this, it is worth mentioning that the operations in our MOM are performed without gradients; in other words, the automatic differentiation is turned-off. This can be easily achieved by calling the *detach()* or *stop_gradient* methods in PyTorch or TensorFlow respectively. If gradients are not disabled for the computation of the occlusion masks in our MOM, it is easy for the network to generate MED probability volumes that give rise to "unreal occlusions", minimizing most of the reconstruction errors in Eq. (6) in our main paper, and preventing the FAL-net from learning.

## 3   Detailed Network Architecture

The detailed network architectures of our light, medium, and heavy (A, B, and C, respectively) FAL-net are described in Table 1. Our FAL-net is an auto-encoder with skip connections kind-of-architecture that takes as input a left view $\mathsf{I}_L$ and outputs a $N$-channel disparity logit volume $\mathsf{D}_L^L$. Inspired by the success of residual connections [12], we adopt residual blocks after each strided convolution in the encoder side. The residual blocks (Resi-Blocks) in our FAL-netB(and C) and FAL-netA are depicted in Figure 1-(a) and (b) respectively. The light version of our FAL-net, the 6.6M parameters FAL-netA, incorporates separable convolutions to reduce its number of parameters further. It can be noted in Table 1 that in the decoder side of our FAL-net, instead of the traditional $2\times$ nearest up-scaling, we use nearest interpolation to the shape of the corresponding encoder skip connection. This makes it easier to test our FAL-net on any input image size without the need for cropping feature map borders, which could cause feature map misalignments during test time. In

a) Resi-Block      b) FAL-netA Resi-Block

Figure 1: Residual blocks in our FAL-netB(and C) and FAL-netA respectively.

Table 1: Detailed network architecture. Conv2d: 3x3 Convolutional layer with a stride of 1, otherwise specified. s2: Stride of 2. Nearest: Nearest upsampling to the size of the skip connection. ELU: Exponential Linear Unit. Ch. A B C: number of channels in FAL-netA, B, and C respectively

| Output | Layer description | Input | Ch. A | Ch. B | Ch. C | Size |
|---|---|---|---|---|---|---|
| $\mathbf{I}_L$ | Input image | - | 3 | 3 | 3 | H×W |
| Conv0 | Conv2d + ELU + Resi-Block | $\mathbf{I}_L$ | 32 | 32 | 32 | H×W |
| Conv1 | Conv2d(s2) + ELU + Resi-Block | Conv0 | 64 | 64 | 64 | H//2×W//2 |
| Conv2 | Conv2d(s2) + ELU + Resi-Block | Conv1 | 128 | 128 | 128 | H//4×W//4 |
| Conv3 | Conv2d(s2) + ELU + Resi-Block | Conv2 | 128 | 256 | 256 | H//8×W//8 |
| Conv4 | Conv2d(s2) + ELU + Resi-Block | Conv3 | 256 | 256 | 256 | H//16×W//16 |
| Conv5 | Conv2d(s2) + ELU + Resi-Block | Conv4 | 256 | 256 | 512 | H//32×W//32 |
| Conv6 | Conv2d(s2) + ELU + Resi-Block | Conv5 | 256 | 512 | 512 | H//64×W//64 |
| Dec6 | Nearest + Conv2d + ELU | Conv6 | 128 | 256 | 256 | Conv5 |
| iConv6 | Concat + Conv2d + ELU | Dec6, Conv5 | 256 | 256 | 512 | Conv5 |
| Dec5 | Nearest + Conv2d + ELU | iConv6 | 128 | 128 | 256 | Conv4 |
| iConv5 | Concat + Conv2d + ELU | Dec5, Conv4 | 256 | 256 | 256 | Conv4 |
| Dec4 | Nearest + Conv2d + ELU | iConv5 | 128 | 128 | 128 | Conv3 |
| iConv4 | Concat + Conv2d + ELU | Dec4, Conv3 | 128 | 256 | 256 | Conv3 |
| Dec3 | Nearest + Conv2d + ELU | iConv4 | 64 | 128 | 128 | Conv2 |
| iConv3 | Concat + Conv2d + ELU | Dec3, Conv2 | 128 | 128 | 128 | Conv2 |
| Dec2 | Nearest + Conv2d + ELU | iConv3 | 64 | 64 | 64 | Conv1 |
| iConv2 | Concat + Conv2d + ELU | Dec2, Conv1 | 64 | 64 | 64 | Conv1 |
| Dec1 | Nearest + Conv2d + ELU | iConv2 | 64 | 64 | 64 | Conv0 |
| $\mathbf{D}_L^L$ | Concat + Conv2d | Dec1, Conv0 | $N$ | $N$ | $N$ | $\mathbf{I}_L$ |

contrast with most previous works, our FAL-net does not provide a multi-scale output or outputs at multiple decoder stages. We observed better performance with only one output at the target resolution.

## 4 Multi-Scale Post-Processing

We define a multi-scale post-processing (PP) step that is less expensive than the PP step proposed in [6]. Our PP involves running the network twice, once for the original image, and a second time on a flip-downscaled image, whose output is unflipped and up-scaled back to target resolution via bilinear interpolation. Processing the same input at a lower resolution allows the receptive field of the FAL-net to cover a larger area, which often results in slightly better predictions for the very close-by objects. However, the processing of a downscaled input also results in a lower quality of depth predictions for far-away objects. To overcome this, instead of naive element-wise averaging between the normal $\mathbf{D}'_L$ and the flip-downscaled $\mathbf{D}_L^{fd}$ depth estimates, we blend them weighted by the normalized disparity $\tilde{\mathbf{D}}_L$. The final post-processed depth $\mathbf{D}_L^*$ is given by

$$\mathbf{D}_L^* = (1 - \tilde{\mathbf{D}}_L) \odot \mathbf{D}'_L + \tilde{\mathbf{D}}_L \odot \mathbf{D}_L^{fd} \tag{1}$$

where the normalized disparity $\tilde{\mathbf{D}}_L$ is given by

$$\tilde{\mathbf{D}}_L = \max \frac{\mathbf{D}'_L}{\text{ptile}(\mathbf{D}'_L, 95)}, 1 \tag{2}$$

| Input image | FAL-netA33 | FAL-netA33 (PP) | FAL-netB33 | FAL-netB33 (PP) |

Figure 2: Effects of our depth-guided multi-scale post-processing (PP) step.

where $\text{ptile}(\mathbf{D}'_L, 95)$ returns the 95th percentile. The use of the percentile instead of the plain max value is adopted to reduce the effects of outliers and noise in the depth estimates. The flip-downscaled disparity is obtained from a 2/3 resolution input image, which is approximately 60% less computationally expensive than running the network on a full-resolution input image as in previous works that incorporate [6]'s post-processing. A qualitative comparison of our models with and without our multi-scale post-processing step is shown in Figure 2 for the FAL-netA33 and the FAL-netB33. A close look at the depth estimates shows that our proposed PP helps in alleviating the discretization artifacts present in homogeneous areas, providing smoothness without degrading the quality of the thin objects and structures.

## 5  Inference times and number of parameters

Table 2 shows a comparison of the number of parameters, inference times on a full-resolution KITTI image, and SqRel between our FAL-netA33, B33, B49, and the previous state-of-the-art (SOTA). Our FAL-netA33 and FAL-netB33 achieve the fastest inference times with 15ms on a full-resolution KITTI image and relatively low SqRel. Our FAL-netB33 already outperforms previous methods in terms of SqRel. Our FAL-netB49 achieves the lowest SqRel, and still, $3\times$ faster inference times than the recent SOTA of PackNet [10]. The robust SqRel metric penalizes more relatively larger errors. Even when the inference times of [5, 16] are not provided in their respective papers, we can assume they are similar to our networks, due to their relatively small network sizes. However, the SqRel errors of [5, 16] are considerably higher than the SqRel of our FAL-netB33 and B49. It is worth noting that the increase in the number of parameters from FAL-netA to the FAL-netB seems not to contribute in the inference time, at least for the TitanXP GPU on which it was measured.

## 6  Multi-Dataset Training

Similar to [8, 9], we perform multi-dataset training with the KITTI [4] and the CityScapes [1] datasets concurrently. We adopt the technique in [8], which sets the maximum disparity hyper-parameter for the CityScapes dataset proportional to the KITI maximum disparity, multiplied by the camera baseline ratio between the CityScapes and the KITTI datasets. The KITTI dataset has a distance between cameras (baseline) of 54cm approx. The CityScapes has a baseline of 22cm. Then, the minimum and maximum disparity hyper-parameters used for the CityScapes images are given by

$$d^{cs}_{min} = 2\frac{22}{54} \tag{3}$$

$$d^{cs}_{max} = 300\frac{22}{54} \tag{4}$$

Table 2: Comparison of our proposed method versus the recent SOTA single image depth estimators in terms of inference times, numbers of parameters, and SqRel (relative squared error from metrics in [2]) on the original [2] and improved [15] KITTI Eigen Test Split. Inference times in milliseconds for a full-resolution KITTI image, otherwise specified

| Method | #Par | Time [ms] | SqRel Original [2] | SqRel Improved [15] | Device |
|---|---|---|---|---|---|
| Monodepth2 [5] | 14 | - | 0.802 | 0.536 | - |
| Monodepth2-s [5] | 14 | - | - | 0.503 | - |
| DepthHints [16] | 35 | - | 0.710 | 0.364 | - |
| Guizilini *et al.* [11] | ∼140 | ∼60 | 0.761 | - | TitanV100 |
| PackNet [10] | 120 | 60 | 0.758 | 0.359 | TitanV100 |
| Tosi *et al.* [14] | 43 | 160 | 0.673 | - | TitanX |
| DORN [3] | 51 | 500 | - | 0.307 | GPU 2.5 Ghz |
| FAL-netA33 | **6.6** | **11** | 0.723 | 0.367 | TitanXP |
| FAL-netB33 | 17 | **11** | 0.633 | 0.304 | TitanXP |
| FAL-netB49 | 17 | 19 | **0.562** | **0.287** | TitanXP |

| Input image | FAL-netA33 (K) | FAL-netB33 w/o MOM (K) | FAL-netB33 (K) | FAL-netB49 (K) | FAL-netB49 (K+CS) |

Figure 3: Additional results on ablation studies.

where 2 and 300 are the minimum and maximum disparity hyper-parameters used for the KITTI dataset, respectively. It is shown in Table 1 of our main paper that the concurrent K+CS training achieves much better performance than the widely used CS pre-training in [6].

# 7   Additional Results

We provide additional results for the ablation studies, the KITTI [4] dataset, the CityScapes [1] dataset, and the Make3D [13] dataset.

## 7.1   Additional Results on Ablation Studies

Figure 3 depicts the depth estimates from our FAL-net trained with various settings (please see Table 1 of our main paper). Quantitatively, the lightweight FAL-netA33 with 6.6M parameters under-performs the heavier FAL-netB49 model. However, the FAL-netA33 still manages to provide decent enough depth estimates, preserving most thin structures. As expected, the models with more quantization levels (49) exhibit fewer discretization artifacts that the sparser models with 33 planes. Overall, as also shown in Tables 1 and 2 of our main paper, our FAL-netB49 trained with KITTI + CityScapes (K+CS) concurrently performs the best among all existing methods.

### 7.1.1   Results without Perceptual Loss

The effect of disabling the perceptual component of our reconstruction loss ($\alpha_p = 0$ in Eq. (6) in our main paper) is shown in Figure 4. Even when the depth estimates of the FAL-netB33 without perceptual loss are still plausible, they suffer from discontinuities in the homogeneous regions, such

|  | | | | |
|---|---|---|---|---|
| Input image | FAL-netB33 (no perceptual) | FAL-netB33 | FAL-netB49 (CS) | FAL-netB49 (K) |

Figure 4: (Left) Additional results of our model trained without perceptual loss. (Right) Additional results of our model trained on CityScapes [1] and evaluated on the KITTI Eigen test split [2]

as roads and buildings. This is because the pure $l_1$ loss does not penalize the relationships between neighboring pixels, which allows the depth discontinuities to minimize the photometric reconstruction errors.

## 7.2 Additional Results on the Eigen Test Split

Additional qualitative comparisons between our FAL-netB49 and the recent state-of-the-art methods of PackNet [10] and DepthHints [16] are presented in Figure 5. It is shown that our FAL-netB49 performs estimates more consistently on thin and complex structures. It is worth noting that our FAL-netB49 has half the number of parameters of DepthHints [16] and seven times fewer parameters than PackNet [10]. Additionally, our proposed method does not require the use of semi-global matching as in DepthHints [16].

## 7.3 Additional Results on CityScapes

Additional results on CityScapes [1] are presented in this Supplemental. Firstly, on the right side of Figure 4 we show that our model trained on CityScapes (CS) and evaluated on the KITTI Eigen test split [2] generalizes well. It can be noted that the FAL-netB49 (CS) generates very plausible depths, perceptually very similar to the FAL-netB49 trained on the KITTI Eigen train split [2] (K). Secondly, in Figure 6, we show depth estimates of our method trained with and without the CityScapes [1] dataset and evaluated on the *eval* folder of CityScapes, which was excluded from the training data. Our models trained without CS generate plausible depth estimates when fed with 60% re-scaled CS images, which is reasonable as the resolution of CityScapes is larger than the KITTI's. Our methods trained with CS, and concurrent K+CS are fed with full-resolution CityScapes images and generate very realistic depths. Moreover, our method trained with K+CS shows that it can generalize the best not only on KITTI, as shown in Table 2 of our main paper, but also on CS, suggesting that the concurrent K+CS training not only improves the performance on KITTI but also on CityScapes.

## 7.4 Additional Results on Make3D

Results on the Make3D [13] dataset are depicted in Figure 7 on a center crop following the protocol in [6]. Among the methods of [6, 7] and ours, our FAL-netB49 (K+CS) shows the closer-to-the-ground-truth estimates (please refer to Table 3 of our main paper for the quantitative comparison). Note that the ground-truths provided in Make3D are of very low-resolution and not aligned precisely with the input images. Interestingly, the FAL-netB49 (K+CS) estimates (in particular those in rows 1, 2, and 8) appear to be a combination of the result presented by the FAL-netB49 (K) and the FAL-netB49 (CS), which is the desired effect when training with additional data.

|  | | | |
|---|---|---|---|
| Input image | Our FAL-netB49 | PackNet [10] | DepthHints [16] |

Figure 5: Qualitative comparison versus the resent state-of-the-art methods on the KITTI Eigen Test Split [2].

Figure 6: Qualitative comparison of our method trained with and without the CityScapes [1] dataset on the *eval* directory of the CityScapes dataset.

Figure 7: Qualitative comparisons on the Make3D [13] dataset.