[Reviews · NeurIPS 2020]

Review 1

Summary and Contributions: This paper proposes a new deep network (FALNet) and (stereo) self-supervised learning algorithm for monocular depth estimation. The network is a lightweight U-Net type with skip connections, similar to the standard "monodepth2" architecture from Godard et al [6]. The output parametrization is not (inverse) depth, as common in state-of-the-art self-supervised approaches [6,11], but averaged softmax votes for exponentially discretized disparity bins (similar to the supervised method DORN [3]). The main contribution is not in the architecture though, but in the stereo-based, occlusion-aware, two-stage self-supervised learning algorithm. First the network is trained using the standard [7] left-right photometric consistency loss via view synthesis (predicting disparities and rigidly warping the left pixels onto the right image with known baseline). Second, a copy of this disparity network is kept frozen during a follow-up fine-tuning stage. The frozen network is used to compute occlusion masks by symmetrically summing left and right, predicted and geometrically warped disparities. This is the so-called "Mirrored Occlusion Module" (MOM). These occlusion masks are then used to fine-tune the network by only paying the photometric (and perceptual) loss on the non-occluded (i.e. valid) regions (Eq. 6). An additional mirror loss is used to supervise all the valid pixels in the left image by maximizing the consistency between the predicted disparities for the left image and the flipped disparities predicted for the flipped left image, as commonly done for post-processing, but masked via the left occlusion mask. This occlusion-aware fine-tuning second stage reduces visibly the typical parallax-induced artifacts around object boundaries and significantly improves the performance of the monocular depth estimation model, establishing a new state of the art on the KITTI, Cityscapes, and Make3D benchmarks.

Strengths: S1: Excellent results. First and foremost, the quantitative performance on these 3 benchmarks is impressive (e.g., 0.088 abs.rel on the original Eigen split of KITTI), outperforming other stereo-supervised and even lidar-supervised approaches. This is furthermore achieved with a much lighter weight model than the current state of the art for (monocular) self-supervised networks (16M vs 120M parameters). This is additional strong evidence that lidar supervision is not needed for monocular depth estimation, furthering recent results in that direction [6,11]. S2: Solid experimental protocol. The evaluation itself is very thorough (typical papers only use KITTI) and accurate, avoiding the sadly not uncommon pitfalls of misreporting results for different splits or different forms of (self-)supervision. The ablative analysis is also very good and exhaustive, highlighting the merits of the proposed occlusion-aware fine-tuning state. In particular, the authors also ensure that their improvements are not only due to longer training schedules. S3: Sound geometric principles. The motivation for the main contribution (MOM) is a geometrically principled way to handle parallax issues in self-supervision via view synthesis. All the various design choices are detailed and justified, as well as properly attributed (at least in the main sections if not in the introduction) when not invented by the authors themselves. I also think that Figure 3, although overwhelming at first, is in fact very well done and precisely overviews the whole approach (it just takes a while to parse).

Weaknesses: I have no major concerns, but only remarks and suggestions for improvements. W1: stereo vs monocular (SfM) self-supervision. Although this is unambiguous in the experimental section, the abstract and introduction should clarify that the method is self-supervised from stereo pairs. There is a lot of confusion in the literature, because all monocular methods predict depth from a single image (by definition) but can be trained in different ways: from lidar supervision (full or partial), from stereo pairs (as is the case here), or from videos (a.k.a. structure-from-motion). Some of the authors' critique of related works (e.g., regarding dynamic objects) are only applicable to the SfM self-supervised scenario, as in the case of stereo-based self-supervised learning pairs of images are captured at the same time. Furthermore, the SfM case requires estimating the camera's ego-motion, which vastly complicates the self-supervised learning task (hence why the comparison is not entirely fair in my opinion). This distinction also brings the question of how can the current approach be extended to the SfM case where only monocular videos are available at training time. A discussion of extensions in this setup (how to adapt MOM, robustness in dynamic environments) would be welcome. W2: clarity. The method is not clearly summarized in the abstract and introduction. Instead, a lot of acronyms are introduced (SIDE, MOM, MED, FAL...) without the intuition behind conveyed clearly. I would suggest using less acronyms (they are mostly levels of indirection and regularly re-defined throughout the paper). I would also suggest giving simple high-level explanations for the intuition behind the contributions. For instance, MOM could be simply introduced as multi-view occlusion masks computed from a fixed disparity network to filter invalid regions due to parallax. W3: Qualitative results. Although some predictions are shown in the main paper and supplementary material, it would be great to also see videos and pointclouds to assess the stability and metric alignment of the predictions. The color-coded disparity maps can indeed be misleading (scale invariant) and the subtle difference between the top performing methods difficult to see. W4: Minimal Broad Impact Statement. This is not a big concern, but I feel that it is currently quite short. There are many questions that could be discussed further, for instance: is this approach systematically overestimating distance? This could be much more dangerous than systematic underestimation for typical robotic applications like self-driving cars.

Correctness: To the best of my knowledge, the equations and experimental methodology is correct and well described.

Clarity: As mentioned above, the abstract / introduction could be improved (cf. above for specific suggestions), but the rest of the paper is clear.

Relation to Prior Work: As discussed above, I believe the most important related works and recent state of the art are discussed and acknowledged appropriately. Note that this is a very active area of research though, so it is possible that I am missing some references, despite my best efforts.

Reproducibility: Yes

Additional Feedback: After reading the rebuttal and the other reviews, I maintain my score of 7 (Accept). I would like to thank the authors for their rebuttal, and I believe they can improve the paper as discussed. I think the excellent results, extensive experiments, and clear motivation make this a worthy NeurIPS paper (I would also accept it at CVPR, as it is a very solid Computer Vision paper).


Review 2

Summary and Contributions: This paper presents a method to train single-image depth estimation network in a self-supervised manner. It supervises on the photo-consistency between stereo views. It proposes a novel module called Mirrored Occlusion Module (MOM) to obtain occlusion maps so that the photo-consistency loss is not imposed on occluded regions. It also proposes a two-step training strategy to train better networks. The proposed self-supervised method outperforms even the best SUPERVISED methods on the KITTI benchmark.

Strengths: + The proposal of using "exponential disparity level" as the representation of depth volume is novel and leads a lot of improvement according to Table 1. This representation is something that we can learn from and adopt. + The performance of the method is strong and surprisingly surpasses that of the state-of-the-art SUPVERVISED methods on KITTI.

Weaknesses: - The novelty of the paper seems to be a bit limited. The major claimed contribution is a Mirrored Occlusion Module (MOM) that obtains occlusion maps from depth maps predictions. The occluded areas are then ignored when penalizing the training loss. This idea is not very different from prior works that obtain confidence maps [8, 34]. The only difference is that the occlusion maps now come from predicted depth instead of network prediction, which is nothing surprising. - Inadequate discussion about the MOM and the training schedule. The occlusion maps are obtained from depth PREDICTIONS, which are expected to contain errors. But it is unclear how the proposed method can recover from the mistakes made in the depth predictions to obtain better depth in the second stage. Moreover, if the same training schedule is adopted by previous approaches, would their performance improved as much? - It is unclear if the benefit of the proposed "exponential disparity level" representation is universal, i.e. if it could improve other methods that perform self-supervised single image depth estimation. The argument would be stronger if the paper could plug in this representation to previous approaches and improve performance.

Correctness: Looks correct.

Clarity: Yes.

Relation to Prior Work: Yes

Reproducibility: Yes

Additional Feedback: ================================================Post Rebuttal I still feel uncertain why the network can gradually correct the mistakes it make in the occlusion and depth estimation. The rebuttal suggested that the occlusion map estimation gets better as FAL-net better, but if the occlusion from MED is not accurate to begin with, how would the FAL-net get better? I think the paper can benefit from an in-depth discussion on this issue. Nevertheless, the authors present strong results in the rebuttal to show that the benefit of MED and MOM modules is universal. These are techniques that we can borrow from. I am raising the rating score.


Review 3

Summary and Contributions: This work provides a self-supervised depth estimation network by employing an occlusion reasoning module and firstly performing the task of view synthesis. To improve the training efficiency, the authors propose to use a log scale instead of a linear one to represent depth discretization. The authors show by reasoning occlusion with their mirrored occlusion module, they obtain better depth estimation by removing the occluded regions from the photometric errors.

Strengths: The idea of assisting depth estimation with occlusion reasoning makes sense, and the paper shows a concerete implementation of this idea. It also shows clean boundaries in the estimated depth maps thanks to this occlusion reasoning module. The method provides solid baseline comparisons against the state of the art, and different versions of this method are shown to outperfom the state of the art.

Weaknesses: The idea of using a log scale for more efficient representation of depth values does not appear new to me, given existing volumetric rendering literature. Volumetric rendering techniques often use such log scales for the same reason. Using view synthesis as a proxy task of depth estimation is one of the core ideas presented, but no such results are shown. It wouuld be interesting to see how the success/failure in this proxy task is correlated to errors in the end goal---depth estimation. ==================== Post-rebuttal feedback: Thanks for clarifying on some of my concerns in the rebuttal. I'd like to maintain my rating. ====================

Correctness: Yes.

Clarity: Mostly yes, but the model figure can use better abstraction to hide some implementation details but show more important high-level ideas.

Relation to Prior Work: I am not particularly familiar with literature on depth estimation in autonomous driving.

Reproducibility: Yes

Additional Feedback:


Review 4

Summary and Contributions: This paper proposes a new network and training method for single view depth estimation. The training is performed in a unsupervised manner with stereo sequences. (1) A light weight depth network (FAL-net) with fewer parameter and faster inference speed is proposed. (2) Occlusion is explicitly considered with a mirrored occlusion module. (3) State of the art result in unsupervised depth learning is presented. Note that the result is even better than some existing SOTA supervised methods (e.g. DORN).

Strengths: - Useful network design There are papers about designing network architecture for depth estimation task. This paper proposes a new network with faster inference speed and less memory burden. - Superior result is presented. The final model achieves state of the art result in unsupervised depth estimation. It is even better than DORN, a SOTA supervised method. - Extensive experiments are performed to prove the effectiveness of the proposed components.

Weaknesses: I am not suere if this is considered as a weakness, maybe just there are some important concepts/sections that I missed from the paper. Anyway, I am not quite understand regarding the use of MOM module. (1) I assume the MOM module is trying to reason out the occlusion regions in the image and act as a weighting mask to neglect these regions in loss calculation. (2) In figure 3 and Sec 3.2.1, it is not clear to me how the summation of MED probability distributions form the occlusion mask. Shouldn't the summation result becomes one for all the pixels? In Eqn(4), I am not sure how does the "max" operation work in the equation. It is quite confusing. --- post rebuttal --- The authors have responded to the comments and it looks more clear to me.

Correctness: /

Clarity: Maybe I missed / misunderstand some sections. i couldn't understand some details of the appraoch as mentioned before. I would say it is not clearly written at this stage. --- post rebuttal ---- As mentioned by the authors, I hope the authors can explain the concepts clearly in the final version

Relation to Prior Work: Yes.

Reproducibility: Yes

Additional Feedback: /

[Author Response · NeurIPS 2020]

Table 1: Results on the original Eigen test split [5] for other models with our MED volumes and MOM

| Ref | Methods | PP | Sup | Data | #Par | abs rel↓ | sq rel↓ | rmse↓ | $rmse_{log}$ ↓ | $a^1$ ↑ | $a^2$ ↑ | $a^3$ ↑ |
|---|---|---|---|---|---|---|---|---|---|---|---|---|
| [7] | Monodepth | | S | K | 32 | 0.148 | 1.344 | 5.927 | 0.247 | 0.803 | 0.922 | 0.964 |
| [7] | Monodepth-MED | | S | K | 32 | 0.112 | 0.751 | 4.500 | 0.196 | 0.868 | 0.954 | 0.980 |
| [7] | Monodepth-MED with MOM fine tune | | S | K | 32 | **0.107** | **0.684** | **4.311** | **0.187** | **0.878** | **0.960** | **0.982** |
| [20] | SuperDepth | | S | K | - | 0.112 | 0.875 | 4.958 | 0.207 | 0.852 | 0.947 | 0.977 |
| [20] | SuperDepth-MED | | S | K | 58 | 0.111 | 0.682 | 4.295 | 0.190 | 0.879 | 0.959 | 0.982 |
| [20] | SuperDepth-MED with MOM fine tune | | S | K | 58 | **0.108** | **0.647** | **4.180** | **0.184** | **0.886** | **0.962** | **0.983** |

Firstly, we thank all reviewers for their detailed reviews. Next, we discuss their comments (R:Reviewer, W:Weakness).

R1.W1: We will clarify in abstract and introduction that our method is self-supervised from stereo pairs. When learning
from mono videos, our approach can be easily adopted, since the MOM can be used as long as the network incorporates
a disparity prob volume, and the relative camera position is known or estimated. The camera-pose information can be
integrated into the warping operation $g(\cdot)$ in Eq. (4) to obtain the mirrored occlusions for the corresponding frame pair.
What could be at stake here is the exponential quantization, as inverse depths in SFM are defined up to an unknown and
inconsistent scale. The ambiguous scale could prevent the network from taking advantage of all disparity levels. A
turn-around for this issue is to incorporate velocity supervision, as introduced in PackNet (Guizilini et al., CVPR2020),
or consistent SFM (Tucker and Snavely, CVPR2020) to fully exploit the exponential quantization.

R1.W2: We will follow this suggestion in our paper to convey the main idea clearly and early on the paper.

R1.W3: We will add videos and point-clouds to the supplementary and presentation materials.

R1.W4: Thanks for the advice, it seems that we miss-understood the purpose of the broad impact statement. It will
be augmented accordingly. Regarding under/overestimation, it seems we are on the safer side. We measured this by
computing the mean median scaling factor [34] between the GT and our depth estimates. We obtained a mean scale
factor of 1.016, indicating our network detects objects slightly closer than they are.

R2.W1: We would like to clarify that the occlusion maps are not obtained directly from the depth predictions, but
from the information coming from the MED volumes of the stereo pair during training (note the MOM is fed with left
and right probability volumes $D_L^P$ and $D_R^P$). Confidence masks in previous methods [8, 34] either require additional
networks or depend on the depth model to generate them, limiting their performance. As can be observed in [8, 34],
their generated masks are of low-quality compared to the highly detailed occlusion maps in our MOM.

R2.W2: Please note that the MOM is fed with the MED volumes from the FAL-net under training, not from the fixed
FAL-net model from the first stage. This means that the generated occlusion maps in the MOM will get better as the
depth estimates from the FAL-net under training get better. We also showed that the improvements just did not come
from the additional training schedule, but from the use of the MOM in the ablation section, we further extend on this in
Table 1 and the next question.

R2.W3: Our experiments tell us that the effectiveness of the "exponential disparity level" representation is universal.
For the sake of completeness, we plugged MED and MOM into the Monodepth (Godard et. al CVPR2017) and the
more recent SuperDepth (Pillai et. al, ICRA2019). The latter incorporates ESPCN [24] up-sampling modules in the
decoder stage. Incorporating MED volumes and Fine-tuning with MOM showed steady improvements in both networks.
This is shown in Table 1 of this rebuttal, which further supports the effectiveness of our overall method.

R3.W1: Surprisingly, we are the first to shed light on the effectiveness of exponential quantization of disparity when
learning self-supervised depth via view synthesis. We showed it could improve accuracy from 84% to 93% just by itself.

R3.W2: During training, we monitored the RMSE between the synthetic right view and the GT right view on the
KITTI2015 training split for our information. We observed a synthesis performance around 22 in RMSE, which is
very good considering that recent works on view synthesis achieve 24 in RMSE [9]. However, since the objective of
our method is not view synthesis, discretization and occlusion artifacts are visible in the synthetic images, as naturally
expected. To reflect the reviewer's comment, we will add visuals of synthesized images in the supplementary material.
The sample of synthetic right view in Fig. 3 of our main paper is actually obtained from the network.

R4.W1: The assumption is correct; we can also borrow a simplified explanation from Reviewer 1: the MOM could be
understood as a multi-view occlusion mask generation module. The generated masks are used to filter invalid regions
due to parallax.

R4.W2: The summation does not become one, as the planes of the probability distribution are first warped (shifted) to
the target view by $g(\cdot)$ in Eq. (4). This shifting not only generates "holes", which are the occluded regions but also
areas where the summation is $> 1$. The latter is the reason why the "max" operator is applied to cap the final occlusion
masks $O^L$ and $O^R$ between 0 and 1. We will make sure it is explained more clearly in the final version of the paper.

[Meta-Review · NeurIPS 2020]

Reviewers like the many positive aspects of this paper, including good results, solid experiments, and sound geometric principles. All reviewers recommend accept after rebuttal. AC agrees with this consensus recommendation. In the camera ready version, the authors should revise the paper as discussed in the rebuttal. In particular, the authors should improve the clarity of the presentation. In addition, the authors should discuss the remaining issue raised by R2 regarding why the network can gradually correct the mistakes it make in the occlusion and depth estimation.